# Comparing a Whole Grain Blend with Polished White Rice for Starch Digestibility and Gut Microbiota Fermentation in Diabetic Patients: An In Vitro Study

**DOI:** 10.3390/foods14152557

**Published:** 2025-07-22

**Authors:** Qian Du, Ruisheng Fu, Ming Zhao, Meihong Xu

**Affiliations:** 1Department of Nutrition and Food Hygiene, School of Public Health, Peking University, Beijing 100191, China; duqian@stu.pku.edu.cn (Q.D.); 1127598380@pku.edu.cn (R.F.); 2Institute of Biotechnology and Health, Beijing Academy of Science and Technology, Beijing 100089, China; 3Beijing Key Laboratory of Toxicological Research and Risk Assessment for Food Safety, Peking University, Beijing 100191, China

**Keywords:** whole grains, in vitro digestion, gut fermentation, diabetes, metagenomics

## Abstract

The high glycemic index (GI) of polished white rice (WR) presents challenges for blood glucose control in diabetes. This study investigated the in vitro digestibility of a whole grain blend (WGB, composed of black, red, and brown rice) and its effects on the gut microbiota in elderly diabetic individuals. WGB exhibited lower starch digestibility (69.76 ± 5.71% vs. 73.02 ± 6.16%) and a reduced estimated glycemic index (eGI, 73.43 ± 4.49 vs. 77.55 ± 2.64) than WR, likely due to its higher amylose content. WGB fermentation increased *Bifidobacterium* and *Lactobacillaceae*, reduced pro-inflammatory *Bacteroides fragilis* and *Enterocloster bolteae*, and released more arabinose and xylose. Additionally, WGB yielded higher isobutyrate, while WR contained more glucose and fructose in its structure, leading to increased acetate production and a more acidic environment. Functional analysis revealed that WGB upregulated pathways related to fatty acid elongation and fiber fermentation. These findings suggest WGB as a viable staple food alternative for diabetic patients, offering dual benefits in glycemic control and gut microbiota.

## 1. Introduction

Diabetes mellitus (DM) is one of the top five chronic diseases worldwide, with a steadily increasing burden on both individuals and healthcare facilities. In 2021, over 537 million adults were living with diabetes, and this number was estimated to reach 783 million by 2045 [1]. Several epidemiological surveys have pointed out that the elderly population suffers a higher prevalence and incidence rate of DM [2,3,4]. Since diabetes treatment is more of a long-term control with not only medications but also the involvement of daily lifestyle changes, it is critical that we delve more into the therapeutic aspects of dietary modifications.

Rice (*Oryza sativa* L.), a staple food for over 3 billion people, is predominantly consumed in its refined form, polished white rice (WR), due to its desirable texture and palatability. However, WR is obtained through the extensive milling of brown rice, removing the bran and germ, leaving only the starchy endosperm. This process not only depletes essential nutrients like B vitamins and minerals but also increases the glycemic index (GI) to an alarming level of 73 [5]. Dietary factors marked by high glycemic load and low fiber intake have been identified as a major contributor in diabetes pathogenesis, in which refined grains, such as WR, play a substantial role [6,7]. A high GI diet is linked to the rapid emergence of postprandial glucose spikes, promoting insulin resistance and pancreatic β-cell dysfunction [8], all of which are critical factors in the development of metabolic imbalance and type 2 diabetes (T2D) [9]. In contrast, whole grains, particularly pigmented rice varieties, retain their bran, germ, and endosperm, preserving a rich content of dietary fiber, vitamins, minerals, and bioactive compounds [10,11,12]. These components have been shown to lower postprandial glycemic levels, enhance insulin sensitivity, and reduce systemic inflammation, suggesting notable benefits of whole grains in diabetes management [13,14].

Emerging evidence highlights the pivotal role of gut microbiota in diabetes progression, and distinct microbial alterations have been observed in diabetic patients [15]. Despite variations in study methodologies and target populations, a consistent pattern of dysbiosis has been documented in diabetic individuals, including reduced microbial diversity, a decrease in butyrate-producing bacteria (e.g., *Faecalibacterium prausnitzii*), and an overabundance of potentially pro-inflammatory taxa such as *Bacteroides fragilis* [15,16,17]. This dysbacteriosis impairs the production of short-chain fatty acids (SCFAs), which are closely related to glucose metabolism and energy homeostasis [15,18]; it also disrupts gut barrier integrity and exacerbates systemic inflammation, thereby contributing to diabetes pathogenesis [19,20]. These findings underscore a bidirectional relationship, wherein gut microbiota dysbiosis is both a consequence of and a contributor to the progression of diabetes.

Although previous studies have explored individual whole grains for glycemic control [12,21,22,23], no research has examined the combined effects of whole grain blends (WGBs) on starch digestibility and gut microbiota modulation. Given the distinct nutrient profiles and functional properties of different whole grains, WGBs may provide enhanced metabolic benefits beyond those of single grains. Additionally, the impact of WGBs on gut microbial metabolism and SCFA production in diabetic populations, particularly the elderly, remains unclear.

Therefore, this study aimed to systematically evaluate the starch digestibility of a WGB (a blend of brown, black, and red rice) using the INFOGEST 2.0 standardized digestion model and employed fecal samples from elderly diabetic patients to simulate disease-specific gut responses to WGB intervention. The findings could provide novel insights into the potential of WGB as a functional staple food for diabetes management.

## 2. Experimental

### 2.1. Materials

The WGB consisted of equal proportions (1:1:1) of black, red, and brown rice, which was designed to reflect common whole grain consumption patterns, while WR served as the control. All rice samples were commercially sourced from a local supermarket in Beijing, China. Both were produced by Jinlongyu, with WR originating from Jilin and WGB from Heilongjiang.

Enzymes including α-amylase (75 U/mL), pepsin (2000 U/mL), and porcine pancreatin (100 U/mL) were purchased from Xiao Dong Pro-Health (Suzhou) Instrumentation Co., Ltd., Suzhou, China. The glucose oxidase–peroxidase (GOPOD) assay kit, total starch assay kit (anthrone colorimetry), and amylose assay kit (iodine colorimetry) were obtained from Suzhou Grace Biological Technology Co., Ltd. (Suzhou, China). All other chemicals used were of analytical grade.

### 2.2. In Vitro Simulated Digestion

#### 2.2.1. Sample Preparation

Rice samples were thoroughly washed and cooked using an electric rice cooker (SY-60YC8001Q, Supor, Hangzhou, China) at a rice-to-water ratio of 1:1.6 (wt/vol) for 35 min. The total starch and amylose contents of the cooked samples were determined using the respective assay kits. Briefly, total starch was quantified using an anthrone-based method. Samples were washed with distilled water, centrifuged (10,000 rpm × 5 min), freeze-dried, and powdered. After extraction with Reagent I (50 °C, 30 min), samples underwent gelatinization (95 °C, 15 min) and acid hydrolysis (HCl, 15 min). The supernatant reacted with a working solution (95 °C, 10 min) before absorbance measurement at 620 nm with standard curve calibration. For amylose content, defatting was performed with 85% ethanol (50 °C, 30 min) and a petroleum ether wash. After gelatinization with Reagent I (95 °C, 10 min), samples were diluted to 10 mL, mixed with Reagents II/III, incubated in the dark (10 min), and measured at 620 nm.

#### 2.2.2. Static Digestion Procedure

The in vitro static digestion consisted of sequential oral, gastric, and intestinal phases. Simulated salivary fluid (SSF), simulated gastric fluid (SGF), and simulated intestinal fluid (SIF) were prepared as described in the INFOGEST protocol [24].

For the oral phase, 5 g of the cooked sample was mixed with 5 mL of SSF containing α-amylase (preheated to 37 °C, pH adjusted to 7.0 with HCl). The mixture was incubated at 37 °C for 2 min under continuous shaking to simulate mastication.

For the gastric phase, 10 mL of SGF containing pepsin (preheated to 37 °C, pH adjusted to 3.0 with HCl) was added, and the sample was incubated at 37 °C for 120 min in an incubator shaker (IS-RDD3, Crystal, New Hyde Park, NY, USA).

For the intestinal phase, 20 mL of SIF containing pancreatin (preheated to 37 °C, pH adjusted to 7.0 with HCl) was added to the gastric chyme. Bile salts were omitted because the rice samples were predominantly composed of carbohydrates with negligible fat content. As bile salts primarily facilitate lipid emulsification and absorption, their inclusion was considered unnecessary in the starch-focused digestion model. Additionally, their presence could potentially introduce interference or variability in the system. The pH was adjusted to 7.0 using 1 M NaOH, and the mixture was incubated at 37 °C for another 120 min. Digestion was performed in triplicate.

During gastric and intestinal phases, 0.5 mL aliquots were collected every 30 min, mixed with 2 mL of anhydrous ethanol to terminate enzymatic activity, and centrifuged at 12,000 rpm, 4 °C, for 10 min. The glucose content at different time points was determined using the GOPOD kit.

#### 2.2.3. Starch Hydrolysis and Glycemic Index Estimation

The starch hydrolysis rate (*S_H_*%) was calculated using the following equation:*S_H_* (%) = *G_p_*/*S_i_* × 0.9
where *G_p_* is the amount of glucose released from starch hydrolysis, *S_i_* is the total starch content in the sample, and 0.9 is the conversion factor from glucose to starch.

The estimated glycemic index (eGI) was calculated using white bread as the reference:*eGI* = 39.71 + 0.549 × *iAUC_test_*/*iAUC_ref_* × 100
where *iAUC_test_* and *iAUC_ref_* are the incremental areas under the digestion curves for the test and reference samples, respectively, computed using GraphPad Prism 10.1.2.

#### 2.2.4. Dialysis of Digested Products

The digested products were heated at 100 °C for 5 min to inactivate enzymes. To remove small-molecular-weight digestion byproducts and collect undigested polysaccharides, the digested samples underwent dialysis using a regenerated cellulose dialysis membrane (molecular weight cut-off: 3.5 kDa, Shanghai Yuanye, Shanghai, China). The dialysis membrane was cut into 15 cm segments, filled with digestion products, and then sealed at both ends. The membrane was immersed in distilled water and dialyzed at 4 °C for 72 h, with the water changed every 12 h. After dialysis, the retentate was collected, freeze-dried (FD-1A-50, Tianjin Biocool Instrument Co., Ltd., Tianjin, China), and ground into powder for further analysis.

### 2.3. In Vitro Fecal Fermentation

#### 2.3.1. Participant Recruitment and Sample Collection

Twenty volunteers with diabetes participated in this study (55% female, 45% male; mean age: 68.40 ± 5.16 years). All participants were free of gastrointestinal diseases and had not taken antibiotics in the past 3 months. The study protocol was approved by the Shougang Hospital of Peking University Medical Ethics Committee (Approval No. IRBK-2024-017-01), and written informed consent was obtained from all participants before study enrollment.

Fresh stool samples were collected in the morning and transported to the laboratory within 4 h. Then, the samples were homogenized for 10 min with phosphate-buffered saline (PBS) at a 1:9 (wt/vol) ratio using a vortex mixer. The homogenates were filtered through sterile gauze and mixed with a yeast-extract–casein hydrolyzed fatty acid (YCFA) medium only or mixed with a YCFA medium containing 0.5% (*w*/*v*) WR or WGB at a 1:10 ratio.

#### 2.3.2. Fermentation Procedure

The freeze-dried digestion products of both WGB and WR were used as substrates in anaerobic culture media (Hangzhou Hailu Medical Technology Co., Ltd., Hangzhou, China), with a blank control containing only digestive fluids.

To establish anaerobic conditions, sealed serum vials were flushed with high-purity nitrogen gas (99.99%) prior to incubation, effectively displacing oxygen. Fermentation was then carried out at 37 °C for 24 h. After incubation, samples were centrifuged at 12,000 rpm, 4 °C for 10 min. The pH of the supernatant was measured using a pH meter (SevenDirect SD20, METTLER TOLEDO, Greifensee, Switzerland). All samples were stored at −80 °C for further analysis.

### 2.4. Short-Chain Fatty Acid (SCFA) Analysis

SCFA analysis was conducted according to the method described by Li et al. [25] with necessary modifications. The supernatant was mixed with NaH_2_PO_3_, incubated at −20 °C for 24 h, and then centrifuged at 13,000 rpm at 4 °C for 20 min. The resulting supernatant was filtered through a 0.22 μm membrane before analysis. SCFA concentrations (acetate, propionate, butyrate, and valerate) were measured using gas chromatography (GC-2010 Plus, Shimadzu, Kyoto, Japan) equipped with a DB-FFAP column (30 m × 0.32 mm × 0.5 mm, Agilent, Santa Clara, CA, USA) and a flame ionization detector. SCFA quantification was conducted using peak integration in GC Solution software Ver. 2.4x, with succinic acid as the internal standard.

### 2.5. Non-Starch Polysaccharides (NSPs) Degradation Products Quantification

To assess NSP degradation, monosaccharide composition analysis was performed following the protocol from Smith et al. [26] with modifications. Briefly, 10 mg of the freeze-dried fermentation residue was hydrolyzed in 0.3 mL of 12 M sulfuric acid at 30 °C for 1 h. The mixture was then diluted with 7.7 mL ultrapure water and further hydrolyzed at 121 °C for 1 h.

The hydrolysates were analyzed using an ion chromatograph (Thermo Fisher 5000+, Waltham, MA, USA), equipped with CarboPac PG1 (4 × 50 mm) and CarboPac PA1 (4 × 250 mm) columns (Dionex, Sunnyvale, CA, USA). Monosaccharide residues were quantified using the response factors relative to fucose as an internal standard.

### 2.6. Metagenomic Analysis of Gut Microbiota

Microbial DNA was extracted from the fermentation products using the QIAamp ^®^ DNA Stool Mini Kit (QIAGEN, Hilden, Germany) following the manufacturer’s instructions. DNA quality was assessed using agarose gel electrophoresis, while concentration and purity were measured using Nanodrop (Thermo Fisher, Waltham, MA, USA) and Qubit fluorometry.

Metagenomic sequencing was performed on the Illumina HiSeq platform (Beijing Microread Genetics Co., Ltd., Beijing, China).

Taxonomic profiling was conducted using Kraken2 with the RefSeq database (NCBI) as a reference. Functional annotation was performed based on the Kyoto Encyclopedia of Genes and Genomes (KEGG) and Carbohydrate-Active enZYmes (CAZymes) databases.

### 2.7. Statistical Analysis

Statistical analyses were performed using R packages (version 4.3.2). Data were expressed as the mean ± standard deviation (SD). Repeated measures ANOVA analysis was used to compare starch hydrolysis rates over time, while differences in starch content and eGI between groups were assessed using independent sample t-tests. Paired t-tests or Wilcoxon signed-rank tests were used for microbial abundance, quantifications of SCFAs and NSP degradation products, depending on normality (Shapiro–Wilk test). LEfSe analysis (LDA threshold = 2.5, Kruskal–Wallis test) was used to identify differentially abundant taxa. The difference was deemed statistically significant with *p* < 0.05.

## 3. Results and Discussion

### 3.1. WGB Had Higher Amylose Content and Reduced Starch Digestibility and Glycemic Response Compared to WR

The total starch content of WGB was significantly higher than that of WR (46.45 ± 2.77% vs. 39.44 ± 0.32%), with a markedly higher amylose content (33.40 ± 1.47% vs. 16.71 ± 3.70%) (Figure 1A). Higher amylose content is associated with reduced starch digestibility, as its tightly packed helical structure limits enzyme accessibility [27]. Therefore, WGB may have a lower starch digestibility due to its higher amylose content compared to WR.

To assess starch digestibility, we employed the INFOGEST 2.0 digestion protocol, which simulates the oral, gastric, and intestinal phases of human digestion using standardized enzymatic conditions [24,28]. This method has been widely validated for studying carbohydrate digestibility and its correlation with in vivo glycemic response [29,30]. Repeated measures ANOVA analysis indicated a notable increase in starch hydrolyzation (*F* = 124.02, *p* < 0.001) in the experimental procedures. Both groups exhibited similar hydrolysis rate trends (*F* = 0.679, *p* = 0.688). Starch hydrolysis remained minimal during the oral and gastric phases for both groups but significantly increased in the intestinal phase, which is compatible with prior studies [31]. This pattern was attributed to the short oral phase, where α-amylase activity was limited, and the absence of starch-digesting enzymes in the gastric phase [32]. WGB exhibited a slightly lower final hydrolysis rate than WR (69.76 ± 5.71% vs. 73.02 ± 6.16%), though the difference was not statistically significant (*F* = 3.32, *p* = 0.078) (Figure 1B, Appendix A).

The eGI values for WGB and WR were 73.43 ± 4.49 and 77.55 ± 2.64, respectively (Figure 1C). Although both values fall within the high-GI category (GI ≥ 70) [5], WGB showed a modestly lower eGI than WR. While the difference was not statistically significant (*t* = 1.367, *p* = 0.243), this moderate reduction may still indicate an improvement in carbohydrate quality, especially considering long-term dietary intake. Notably, these results are consistent with previous findings showing that pigmented rice varieties, such as red and black rice, often display high eGI values despite their enhanced nutritional composition [12].

The observed difference may be related to WGB’s higher amylose content and unique phytochemical profile. Previous studies have shown that amylose-rich foods are digested more slowly, leading to lower glycemic responses [33,34]. Additionally, pigmented rice varieties contain bioactive compounds such as γ-oryzanol, phenolic acids, anthocyanins, and proanthocyanidins [35,36,37,38], which can inhibit α-amylase and α-glucosidase activity, reducing starch digestion rate and postprandial glycemia [12,39].

Collectively, these findings suggest that although WGB does not drastically alter eGI classification, it may offer a more favorable starch digestion profile than WR, potentially contributing to better long-term glycemic control for individuals with impaired glucose metabolism.

### 3.2. WGB Restored Microbial Diversity and Enriched Beneficial Taxa

#### 3.2.1. α-Diversity and β-Diversity Analysis

The Simpson, Evenness, and Shannon indices (Figure 2A–C) indicated that WR significantly reduced α-diversity compared to the blank group (*p* < 0.05), whereas WGB largely preserved microbial diversity. This suggested that WGB helped maintain gut microbial richness, whereas WR consumption may inflict a negative impact on microbial stability.

For the evaluation of β-diversity, distance-based multivariate redundancy analysis (dbRDA) was performed using group variables as explanatory factors and microbial relative abundance as response variables, with covariates of age, sex, and BMI controlled. The dbRDA results (Figure 2D) revealed distinct clustering patterns among groups, with dbRDA1 and dbRDA2 explaining 20.4% and 18.4% of the total variation, respectively. Permutational multivariate analysis of variance (PERMANOVA) analysis further supported the significant group differences in microbiota composition, with an R^2^ of 0.07365. Covariates such as BMI (R^2^ = 0.01748) and age (R^2^ = 0.01579) contributed to the variation but to a lesser extent than group factors, while sex had minimal impact (Figure 2E). Altogether, WGB was helpful in maintaining both α- and β-diversity of the gut microbiome community.

#### 3.2.2. Taxonomic Shifts in Gut Microbiota

At the phylum level, over 95% of the identified bacteria in all groups belonged to *Firmicutes*, *Bacteroidetes*, *Actinobacteria*, and *Proteobacteria* (Figure 3A), which was consistent with previous studies [40], confirming the typical composition of gut microbial communities. The *Firmicutes*-to-*Bacteroidetes* (F/B) ratio, an indicator of gut inflammation in T2D patients [41], was significantly lower in WGB than in WR (*p* < 0.05), suggesting that WGB posed a relatively lower risk of promoting gut inflammation.

At the genus level, WGB and WR significantly elevated the abundance of *Bifidobacterium* and multiple members of *Lactobacillaceae* (Figure 3C,D), which are beneficial SCFA-producing bacteria that inhibit the growth of harmful bacteria, improve immune function [42], modulate oxidative stress and inflammation [43], and support gut barrier integrity [44].

WGB also led to a notable reduction in pro-inflammatory bacteria, including *Bacteroides fragilis* and *Enterocloster bolteae* (Figure 3D). *B. fragilis* is an opportunistic pathogen that has been found linked to T2D [16] and dysbiosis [45]. *E. bolteae* has been shown to be abundantly enriched in patients with T2D [46] and Chronic Hepatitis B [47] and may contribute to metabolic dysregulation by disrupting bile acid metabolism [48]. Since diabetes is often associated with impaired gut barrier function and immune imbalance, a lower abundance of *B. fragilis* and *E. bolteae* may potentially reduce its pathogenic risk and contribute to maintaining gut homeostasis.

While both WR and WGB led to a reduction in butyrate-producing bacteria such as *Faecalibacterium prausnitzii*, *Clostridium*, and *Roseburia* (Figure 3C,D), WGB exhibited a milder reduction, suggesting a lesser impact on butyrate production.

LEFse analysis (LDA score > 2.5) further identified group-specific bacterial enrichments (Figure 3E). In the blank group, the microbial composition exhibited dysbiosis, characterized by an increased abundance of potential pathogens, such as *Clostridium* sp. *M62.1*, *Lacrimispora saccharolytica*, *Clostridioformis*, and *Lachnoclostridium* sp. YL32. Consistent with previous analyses, both WGB and WR groups showed an enrichment of *Lactobacillaceae* and *Bifidobacterium*, indicating a general increase in beneficial taxa compared to the blank group.

These results indicated that WGB contributed to a healthier gut microbiota and enhanced beneficial bacterial populations but reduced inflammation-associated taxa. This microbial shift may partly explain WGB’s metabolic benefits, as gut microbiota alterations are strongly linked to glucose metabolism and insulin sensitivity [15].

### 3.3. WGB Modulated Microbial Metabolism: SCFA Production and NSP Degradation

#### 3.3.1. SCFA Production and Functional Implications

Short-chain fatty acids (SCFAs), primarily acetate, propionate, and butyrate, are crucial microbial metabolites that influence host metabolic health [49]. Variations in SCFA production are largely shaped by gut microbial diversity and substrate availability. Through quantifying the levels of SCFAs in the fermentation products (Figure 4A,B), we were able to investigate the functional consequences of microbial shifts.

Total SCFAs and acetate levels were significantly higher in WR than in WGB (*p* < 0.05) (Figure 4B), which corresponded to the lower pH observed in WR fermentation (Figure 4A). This aligned with WR’s higher starch digestibility and rapid fermentation, leading to greater acetate accumulation and acidification of the fermentation environment. Acetate was the most abundant SCFA, accounting for 66.5% of total SCFAs in WR and 60.4% in WGB. While acetate plays a role in lipid metabolism, excessive levels have been reported to be associated with microbial dysbiosis and inflammatory conditions [50]. Additionally, a highly acidic environment can inhibit the growth of beneficial bacteria [51], suggesting that WR fermentation was more likely to promote a microbial composition that favors pro-inflammatory pathways compared to that of WGB.

Other SCFAs, including propionate, butyrate, and valerate, are present in lower amounts but offer significant health benefits, such as gut barrier maintenance and anti-inflammatory effects [19]. Compared to the Blank group, WGB and WR showed lower levels of propionate, isobutyrate, and isovalerate (Figure 4B). However, the WGB group retained significantly higher isobutyrate levels compared to WR (*p* < 0.05), supportive of the fact that WGB had a more metabolically favorable gut microbiota composition. It is important to note that fecal SCFA concentrations represent only about 5% of the total SCFAs produced in the colon, as the majority are absorbed by colonocytes [52]. Therefore, these results offered insight into microbial fermentation activity rather than actual SCFA production levels.

#### 3.3.2. NSP Degradation and Fermentable Sugar Composition

In order to further investigate the microbial metabolism of rice samples, we analyzed non-starch polysaccharides (NSP) degradation products (Figure 4C–J). The concentrations of arabinose and xylose were significantly higher in fermented WGB than that in WR (Figure 4D,G), while glucose and fructose were more abundant in the WR group (Figure 4F,I), reflecting different fermentation substrates between the two groups. The increased arabinose and xylose may originate from arabinoxylans abundant in black and red rice, which are known to promote the growth of beneficial gut microbiota, particularly *Bifidobacterium* and *Lactobacillaceae* [53]. These microbes ferment arabinoxylan to SCFAs, improving glucose and insulin homeostasis, and additionally regulate key metabolites such as 12α-hydroxylated bile acids, thereby alleviating T2D symptoms [54]. The higher arabinose and xylose levels in WGB fermentation product suggested a slower, fiber-driven fermentation process, which conformed to its lower eGI and higher amylose content. This suggests a prebiotic effect of arabinoxylan-rich grains, potentially contributing to glucose regulation.

In contrast, the WR group exhibited higher glucose and fructose concentrations (Figure 4F,I), implying a greater abundance of simple sugars in the fermentation substrate, which is consistent with the previous result of WR’s rapid fermentation and excessive acetate production [39].

These findings indicate that WR, with its higher concentration of readily fermentable carbohydrates such as glucose and fructose, facilitates rapid fermentation, predominantly producing acetate. Conversely, WGB, which contains more resistant starch and fiber, promotes a slower, fiber-driven fermentation process, potentially leading to better glucose homeostasis and reduced pro-inflammatory effects.

### 3.4. WGB Altered Gut Microbial Function: KEGG and CAZyme Analysis

To explore the functional implications of microbial alterations, we conducted KEGG pathway enrichment analysis and CAZyme profiling (Figure 5).

KEGG analysis revealed significant differences in metabolic pathways among the groups. Compared to the blank group, WGB upregulated 87 metabolic pathways, while WR upregulated 80 (Figure 5A). Fatty acid elongation pathway was significantly enriched in WGB (*p* < 0.05), suggesting enhanced lipid metabolism (Figure 5B). This is interesting because fatty acid elongation is associated with improved insulin sensitivity and reduced inflammation [55]. Both WGB and WR reshaped gut microbiota, leading to changes in host immunity, fatty acid, and bile acid metabolism (Figure 5C,D).

CAZyme analysis identified 37 upregulated families in WGB compared to WR (Figure 5E), mainly belonging to carbohydrate-binding modules (CBMs), glycoside hydrolases (GHs), and polysaccharide lyases (PLs) (Figure 5F). These enzyme families play key roles in dietary fiber degradation. Notably, GH115, GH10, and CBM71 are functionally involved in arabinoxylan metabolism, whose elevation corresponded to higher arabinose and xylose release in the WGB group. Additionally, upregulated GH92, GH117, GH133, and PL11 may be associated with enhanced degradation of complex polysaccharides. These results further supported that WGB promoted a gut microbiota that is inclined to efficiently ferment dietary fiber, leading to improved metabolic outcomes.

## 4. Conclusions

This study utilized a standardized in vitro digestion model based on the INFOGEST protocol and fermentation model with fecal samples from elderly diabetic patients to evaluate the digestibility and gut microbiota-modulating effects of a whole grain blend (WGB, composed of brown, black, and red rice) compared to polished white rice (WR).

We demonstrated that WGB reduced starch hydrolysis and presumably glycemic response, likely due to its higher amylose content and phytochemical composition, potentially beneficial for blood glucose management in diabetic individuals.

In addition, in vitro fecal fermentation revealed that WGB supported a more abundant gut microbiota, enriched beneficial taxa (*Bifidobacterium* and *Lactobacillaceae*), reduced pro-inflammatory taxa (*Bacteroides fragilis* and *Enterocloster bolteae*), and improved the decrease in isobutyrate, while WR promoted acetate-driven dysbiosis and inflammation risk. WGB also enhanced the fermentation of arabinoxylan, leading to higher arabinose and xylose levels, while fermentation product of WR contained more glucose and fructose, in concert with its faster starch breakdown and higher acetate production.

Furthermore, WGB upregulated fatty acid elongation pathways and CAZyme involved in the degradation of complex polysaccharides, which further supported its inclination toward fiber-driven fermentation process.

Collectively, these findings demonstrated that the WGB holds promise as a functional staple food for glycemic control and gut microbiota modulation in diabetic patients, serving as a potential alternative to polished WR. While the current in vitro model cannot fully capture the complexity of host–microbiota–immune interactions, it provides a controlled and disease-relevant platform for mechanistic insights into diet–microbiota–metabolite relationships. Further in vivo studies are warranted to validate these findings, assess long-term metabolic benefits, and confirm their clinical applicability.

## Figures and Tables

**Figure 1 foods-14-02557-f001:**
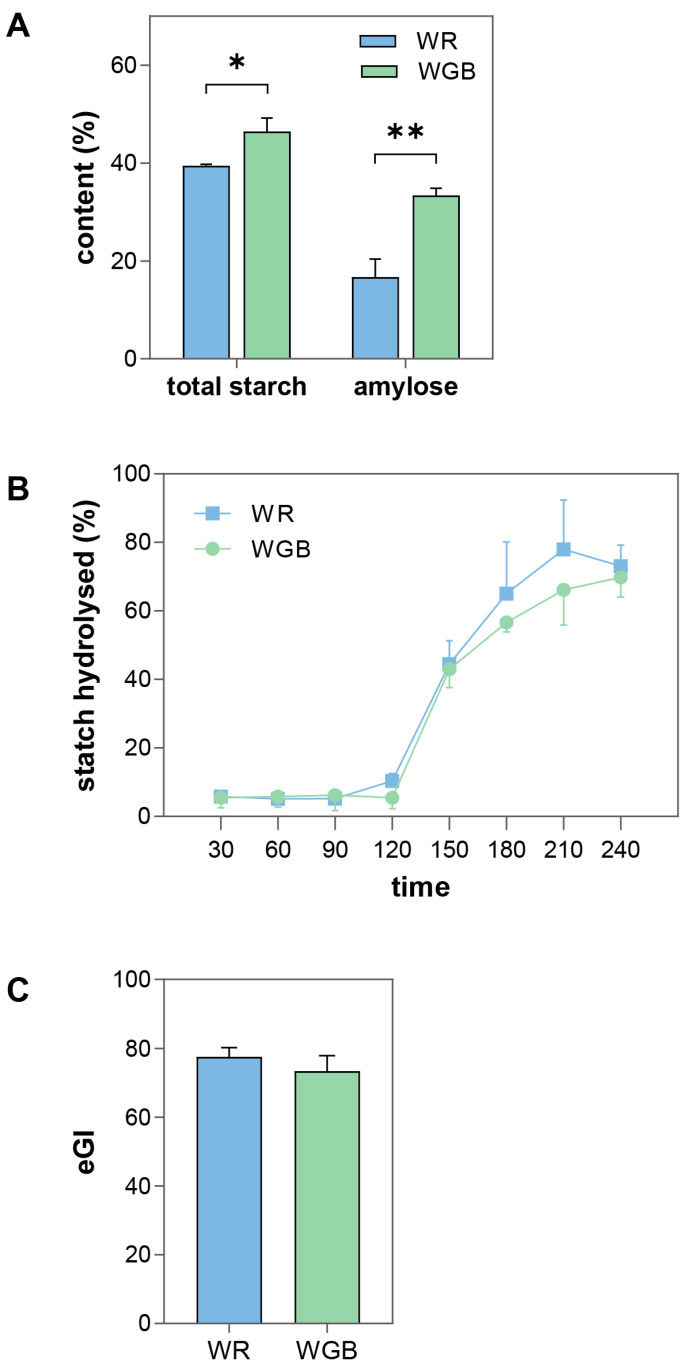
**Starch content, in vitro hydrolysis rate, and estimated glycemic index (eGI) of WR and WGB**. (**A**) Total starch and amylose contents in cooked rice samples. (In (**A**), the labels for WR and WGB were inadvertently reversed in the original version. This error has been corrected in the figure.) (**B**) Starch hydrolysis rate over time during in vitro digestion. (**C**) eGI of the two samples, calculated from the area under the curve for starch hydrolysis. WR: polished white rice; WGB: whole grain blend. Significant differences between groups are indicated by * (*p* < 0.05) and ** (*p* < 0.01).

**Figure 2 foods-14-02557-f002:**
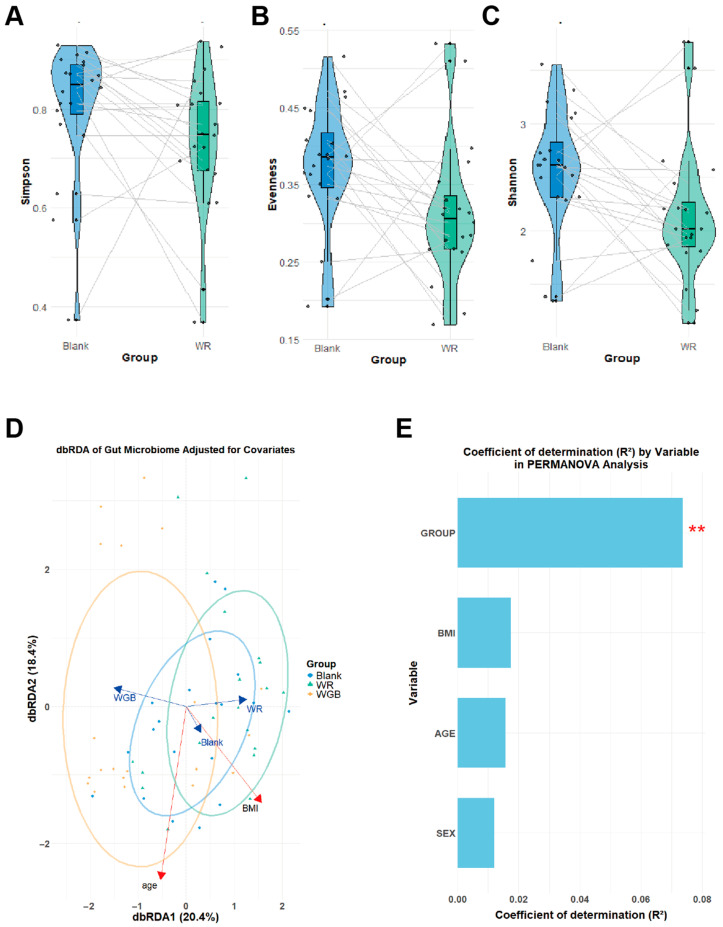
**Changes in gut microbiota composition of samples after in vitro fermentation**. (**A**–**C**) α-diversity indices of the blank and WR groups, including Simpson index (**A**), Evenness index (**B**), and Shannon index (**C**). Each black dot represents an individual data point. (**D**) Distance-based redundancy analysis (dbRDA) illustrating the constitute differences of microbial community between groups. (**E**) Variance contribution of explanatory factors and covariates by the PERMANOVA analysis. Significant differences between groups are indicated by ** (*p* < 0.01). WR: polished white rice; WGB: whole grain blend.

**Figure 3 foods-14-02557-f003:**
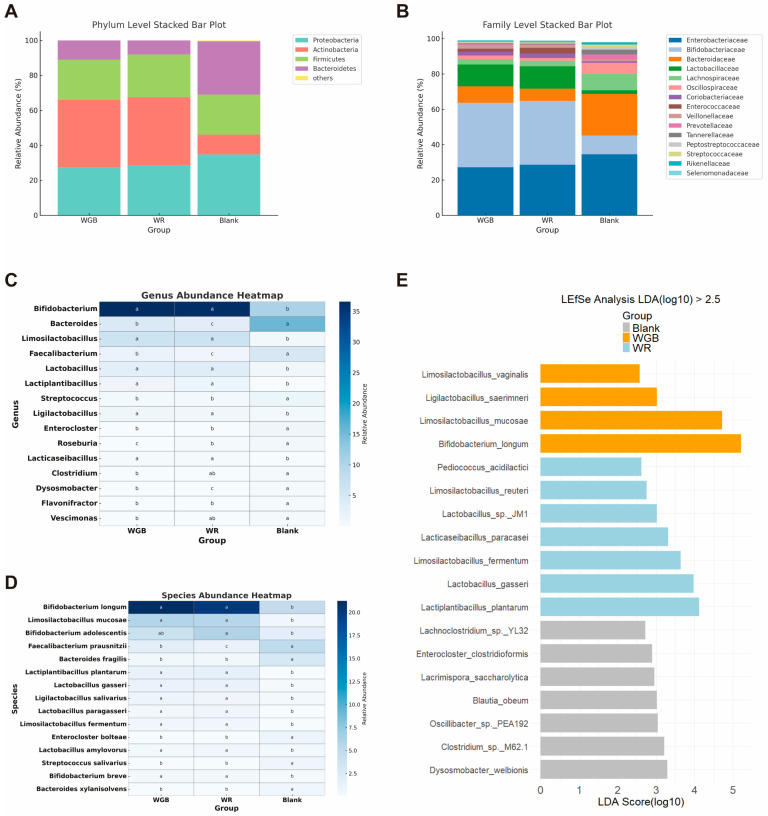
**Changes in microbial composition after fermentation with feces from diabetic patients**. (**A**,**B**) Relative abundance of the top 15 taxa at the phylum (**A**) and family (**B**) levels. (**C**,**D**) Heatmaps showing the top 15 differentially abundant gut microbiota at the genus (**C**) and species (**D**) levels across groups. Different letters indicate statistically significant differences between groups (*p* < 0.05). (**E**) LEfSe analysis results, displaying the LDA (Linear Discriminant Analysis) scores for significantly enriched taxa (log10 > 2.5). WR: polished white rice; WGB: whole grain blend.

**Figure 4 foods-14-02557-f004:**
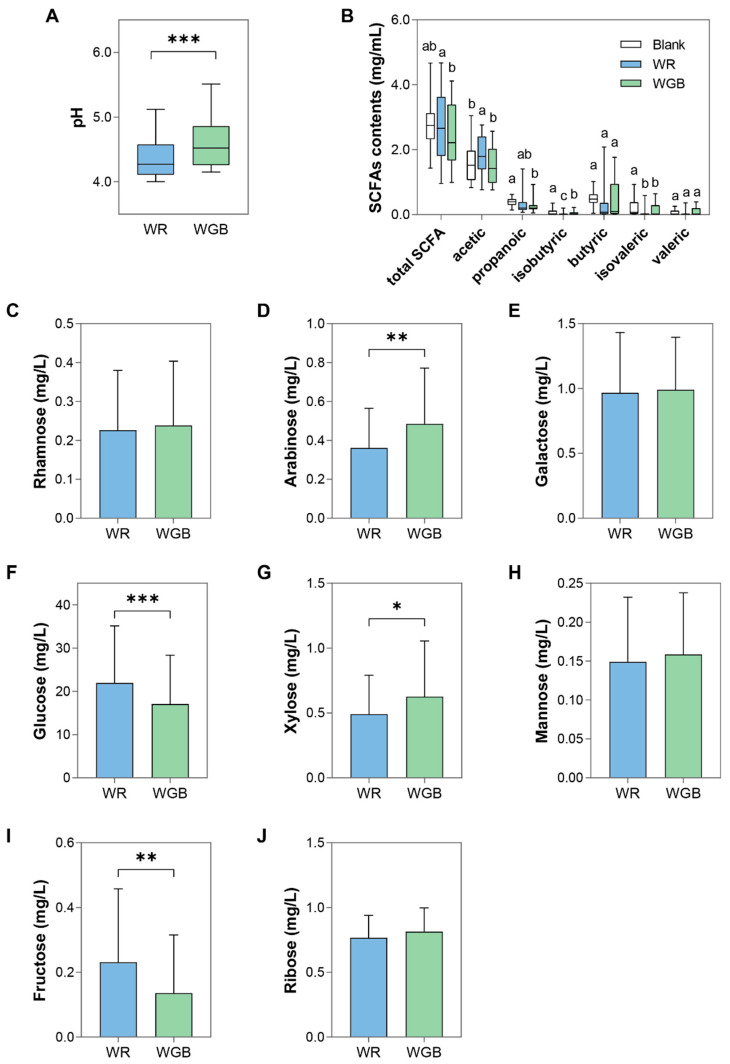
pH, short-chain fatty acid (SCFA) concentrations, and monosaccharide compositions of WR and WGB groups after in vitro fermentation with feces from diabetic patients. (**A**) pH values of fermentation products. (**B**) Short-chain fatty acid (SCFA) concentrations, with different letters indicating statistically significant differences between groups (*p* < 0.05). (**C**–**J**) Monosaccharide compositions derived from the degradation of non-starch polysaccharides (NSP) after digestion, including rhamnose (**C**), arabinose (**D**), galactose (**E**), glucose (**F**), xylose (**G**), mannose (**H**), fructose (**I**), and ribose (**J**). Significant differences in these panels are indicated by * (*p* < 0.05), ** (*p* < 0.01), and *** (*p* < 0.001). WR: polished white rice; WGB: whole grain blend.

**Figure 5 foods-14-02557-f005:**
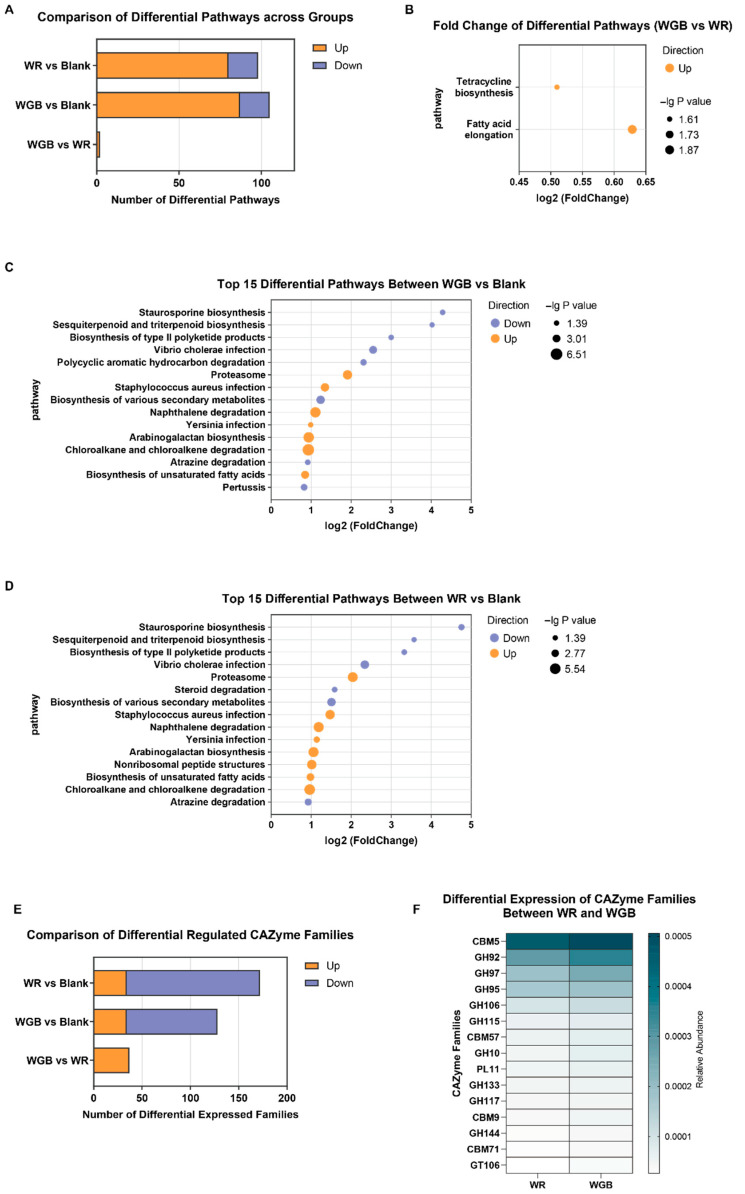
**Functional annotation of gut microbiota in the WR and WGB Groups**. (**A**) The number of differentially enriched pathways in comparisons of WGB vs. blank, WR vs. blank, and WGB vs. WR. (**B**–**D**) Differentially enriched KEGG pathways in WGB vs. WR (**B**), WGB vs. blank (**C**), and WR vs. Blank (**D**). The *x*-axis represents log_2_ (Fold Change), the bubble size corresponds to −log_10_ (*p*-value), and the color indicates the direction of regulation. (**E**) The number of differentially enriched CAZyme families among groups. (**F**) The most significantly expressed CAZyme families between WGB and WR. WR: polished white rice; WGB: whole grain blend.

## Data Availability

The raw metagenomic sequencing data generated in this study have been deposited in the NCBI Sequence Read Archive (SRA) under the BioProject accession number PRJNA1258611. The data are publicly available at: https://www.ncbi.nlm.nih.gov/bioproject/PRJNA1258611.

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
