# Peer review of "Comparing a Whole Grain Blend with Polished White Rice for Starch Digestibility and Gut Microbiota Fermentation in Diabetic Patients: An In Vitro Study"

_foods, 2025, doi:10.3390/foods14152557_

Round 1
Reviewer 1 Report
Comments and Suggestions for Authors
Abstract
The abstract describes the study's main results, which are corroborated by the described methodology. However, authors are encouraged to include specific figures and data on the results explored.
Introduction:
Although the introduction discusses sufficient evidence on the epidemiological burden of diabetes, the effects of foods with a high glycemic index, and polished rice, it lacks sufficient information to highlight the effects of dysbiosis in diabetic patients. The authors are advised to highlight the state of the art regarding the effects of dysbiosis and glucose metabolism.
Methodology
Section 2.2
The methodology described demonstrates scientific rigor.
The authors are only advised to explain or specify why bile salts were not added to the intestinal phase. This is because the static digestion procedure commonly uses bile salts to simulate the intestinal phase.
Section 2.3.2
Authors are advised to specify in detail how they ensured fermentation under anaerobic conditions.
The methodology section does not describe the procedure for determining total starch and amylose content. Authors are encouraged to include a description of these procedures, as they are essential results presented in the Results section and the Abstract.
Results
Lines 199-208: Authors are encouraged to include specific data and figures for the results found. Although statistical data are shown, the scope of the results is not evident.
Lines 212-224: The authors are advised to provide a substantial explanation for their results. While the differences in amylose content (and possibly its digestibility) are interesting, there is no clearly significant difference in the glycemic index between the two samples.
Author Response
Point-by-point response to Reviewer 1
Comments 1: Abstract
The abstract describes the study's main results, which are corroborated by the described methodology. However, authors are encouraged to include specific figures and data on the results explored.
Response 1: Thank you for this valuable suggestion. In response, we have revised the abstract to include key quantitative results regarding starch digestibility (69.76 ±â€¯5.71% vs. 73.02 ±â€¯6.16%) and estimated glycemic index (eGI, 73.43 ±â€¯4.49 vs. 77.55 ±â€¯2.64) of WGB and WR, respectively. These additions are now presented on Lines 17–19 of the revised manuscript.
Comments 2: Introduction
Although the introduction discusses sufficient evidence on the epidemiological burden of diabetes, the effects of foods with a high glycemic index, and polished rice, it lacks sufficient information to highlight the effects of dysbiosis in diabetic patients. The authors are advised to highlight the state of the art regarding the effects of dysbiosis and glucose metabolism.
Response 2: We thank you for emphasizing this important aspect. As noted in our original submission, we discussed the bidirectional relationship between gut microbiota dysbiosis and diabetes progression in the third paragraph of the Introduction section. Specifically, we noted that not only do diabetic patients exhibit altered gut microbiota composition, but these microbial changes in turn contribute to impaired glucose metabolism and systemic inflammation, thereby accelerating disease progression. In accordance with your suggestion, we have now further refined and expanded this section to better highlight the role of dysbiosis in glucose metabolism. The revised description can be found at Lines 54-65 in the updated manuscript.
Comments 3: Section 2.2
The methodology described demonstrates scientific rigor.
The authors are only advised to explain or specify why bile salts were not added to the intestinal phase. This is because the static digestion procedure commonly uses bile salts to simulate the intestinal phase.
Response 3: In our study, the test substrates were cooked rice samples primarily composed of carbohydrates with minimal fat content. Since the main function of bile salts is to emulsify lipids during digestion, their inclusion was deemed unnecessary for simulating the intestinal digestion of starch-based staple foods. Additionally, we found that bile salts tended to precipitate or form micelles in the absence of sufficient lipids, which may introduce variability or interfere with reproducibility. Therefore, to maintain a simplified and targeted in vitro model focused on starch digestibility, we excluded bile salts from the intestinal phase. We have now added this explanation in the revised Methods section (Section 2.2.2, Lines 117-121).
Comments 4: Section 2.3.2
Authors are advised to specify in detail how they ensured fermentation under anaerobic conditions.
Response 4: We confirm that anaerobic conditions were established by flushing the sealed serum vials with high-purity nitrogen gas prior to incubation. This method effectively displaces oxygen and maintains an oxygen-free environment throughout the fermentation process. This clarification has now been added in the revised manuscript at Lines 165-167.
Comments 5: The methodology section does not describe the procedure for determining total starch and amylose content. Authors are encouraged to include a description of these procedures, as they are essential results presented in the Results section and the Abstract.
Response 5: We have added a detailed description of the procedures used to determine total starch and amylose content in Section 2.2.1 (Lines 97-105).
Comments 6: Lines 199-208: Authors are encouraged to include specific data and figures for the results found. Although statistical data are shown, the scope of the results is not evident.
Response 6: We appreciate your point. The text in Lines 199-208 described the dynamic trend of starch hydrolysis during digestion, which is visualized in Figure 1B. The results of repeated-measures ANOVA (analyzing hydrolysis rates at all time points: 30, 60, 90, 120, 180, 210, and 240 min) are reported in text. We hope this supplementary table provides full transparency into the temporal starch hydrolysis trends.
Supplementary Table S1. Starch hydrolysis rates (%) of WR and WGB during in vitro digestion and statistical analysis by repeated measures ANOVA.
|
Time (min) |
WR (mean ± SD, %) |
WGB (mean ± SD, %) |
Group Effect |
Time Effect |
Time × Group interaction effects) |
|
30 |
5.81±1.41 |
5.42±2.90 |
F=3.32 |
F=124.02 |
F=0.679 |
|
60 |
5.13±0.95 |
5.79±3.06 |
P =0.078 |
P < 0.001 |
P =0.688 |
|
90 |
5.24±0.48 |
6.23±4.49 |
|
|
|
|
120 |
10.38±2.12 |
5.44±3.17 |
|
|
|
|
150 |
44.44±6.81 |
42.96±5.35 |
|
|
|
|
180 |
65.05±15.11 |
56.61±2.73 |
|
|
|
|
210 |
78.00±14.38 |
66.10±10.19 |
|
|
|
|
240 |
73.02±6.16 |
69.76±5.71 |
|
|
|
Values are presented as mean ± SD. Statistical significance was assessed using repeated-measures ANOVA with time, group, and interaction effects. WR: polished white rice; WGB: whole grain blend
Comments 7: Lines 212-224: The authors are advised to provide a substantial explanation for their results. While the differences in amylose content (and possibly its digestibility) are interesting, there is no clearly significant difference in the glycemic index between the two samples.
Response 7: We agree that the eGI difference between WR (77.55 ± 2.64) and WGB (73.43 ± 4.49) did not reach statistical significance (t =1.367, P = 0.243), and thus we have revised our wording to more accurately reflect the data. Instead of highlighting WGB as clearly superior, we now emphasize its moderately improved starch digestibility profile and potential metabolic advantage. Additionally, we have clarified that the slight reduction in eGI may still hold physiological relevance, especially when considering chronic dietary patterns and cumulative glycemic burden. Please see the revised discussion at Lines 232–238 and 245-248.
Reviewer 2 Report
Comments and Suggestions for Authors
The research topic addressed in this study is of practical and scientific significance. The manuscript presents insightful findings on the effects of whole grains (including black, red, and brown rice) on gut microbiota in elderly individuals with diabetes. The authors have conducted extensive work and obtained meaningful results that support their hypothesis. However, I have a few suggestions to further improve the quality and clarity of the manuscript:
- The abstract is well-structured; however, it would benefit from including key numerical results to better reflect the magnitude and significance of the findings. Quantitative data can enhance the impact and clarity for readers.
- The sequence and flow of the introduction section could be improved. For example, the second paragraph (starting at line 40) should be placed earlier to establish context more effectively. Specifically, the content in line 38 could be moved before line 45 to maintain a logical progression of ideas.
- Line 133: The manuscript states that 55% of the volunteers were female, but there is no mention of the remaining 45%. It is unclear whether the remaining participants were male or not specified. Clarification on the gender distribution of all participants is needed for a complete understanding of the study population.
- Line No. 212: The reported estimated glycemic index (eGI) value of WGB appears unusually high and raises concerns about accuracy or clarity. It would be helpful if the authors could provide a brief explanation or justification for this value within the results or discussion section. Additionally, to improve transparency and reproducibility, the brand or manufacturer names of the black, red, and brown rice varieties used in the study should be specified in the Materials section.
Author Response
Point-by-point response to Reviewer 2
Comments 1: The abstract is well-structured; however, it would benefit from including key numerical results to better reflect the magnitude and significance of the findings. Quantitative data can enhance the impact and clarity for readers.
Response 1: Thank you for this valuable suggestion. In response, we have revised the abstract to include key quantitative results regarding starch digestibility (69.76 ±â€¯5.71% vs. 73.02 ±â€¯6.16%) and estimated glycemic index (eGI, 73.43 ±â€¯4.49 vs. 77.55 ±â€¯2.64) of WGB and WR, respectively. These additions are now presented on Lines 17–19 of the revised manuscript.
Comments 2: The sequence and flow of the introduction section could be improved. For example, the second paragraph (starting at line 40) should be placed earlier to establish context more effectively. Specifically, the content in line 38 could be moved before line 45 to maintain a logical progression of ideas.
Response 2: We appreciate your insightful suggestion regarding the structure and flow of the Introduction. Originally, our intention was to first present the global burden of diabetes and its impact on the elderly population, followed by the role of dietary factors (such as glycemic load and fiber intake) in diabetes pathogenesis, and then to narrow the focus onto white rice and whole grain alternatives. However, we agree that reordering the content to first introduce rice—our primary research subject—followed by its nutritional limitations (high GI), and then discussing the implications of high-GI diets, provides a more logical and reader-friendly narrative.
Accordingly, we have revised the sequence of the paragraphs as advised: The paragraph introducing rice and its refining process has been moved earlier; The section on dietary glycemic load and refined grains now follows, transitioning naturally into the discussion of the health impact of high-GI diets and the benefits of whole grains, which can be found at Lines 38-45.
To maintain the correct in-text citation order, we have updated the reference numbers throughout the Introduction to reflect the revised sequence.
Comments 3: Line 133: The manuscript states that 55% of the volunteers were female, but there is no mention of the remaining 45%. It is unclear whether the remaining participants were male or not specified. Clarification on the gender distribution of all participants is needed for a complete understanding of the study population.
Response 3: We have clarified the gender distribution of the study population in the revised manuscript. Specifically, we now state that 55% of participants were female and 45% were male (Line 149).
Comments 4: Line No. 212: The reported estimated glycemic index (eGI) value of WGB appears unusually high and raises concerns about accuracy or clarity. It would be helpful if the authors could provide a brief explanation or justification for this value within the results or discussion section.
Response 4: We appreciate your attention to this detail. While the eGI value of WGB (73.43 ± 4.49) appears unexpectedly high for a whole grain sample, it aligns with published studies on pigmented rice varieties. For example, Rebeira et al. [12] reported that several types of red and black rice also exhibited eGI values above 70. These elevated eGI values are partly attributable to the natural variation in starch structure and residual digestibility of pigmented rice, despite their higher content of fiber and bioactive compounds. We have added this explanation in the revised Rusult section (Lines 232–238, 245-248).
Comments 5: Additionally, to improve transparency and reproducibility, the brand or manufacturer names of the black, red, and brown rice varieties used in the study should be specified in the Materials section.
Response 5: As suggested, we have specified the brand and origin of both WR and WGB in the Materials section to improve transparency and reproducibility (Lines 84-85).
Round 2
Reviewer 2 Report
Comments and Suggestions for Authors
Accept.